# Green Leaf Volatile Function in Both the Natural Defense System of *Rumex confertus* and Associated Insects' Behavior

Dariusz Piesik [1,*], Jacek Łyczko [2], Krzysztof Krawczyk [3], Magdalena Gantner [4], Jan Bocianowski [5], Veronika Ruzsanyi [6] and Chris A. Mayhew [6]

1 Department of Biology and Plant Protection, Bydgoszcz University of Science and Technology, 7 Prof. Kaliskiego Ave., 85-796 Bydgoszcz, Poland
2 Department of Food Chemistry and Biocatalysis, Wrocław University of Environmental and Life Sciences, Norwida 25, 53-375 Wrocław, Poland
3 Department of Virology and Bacteriology, Institute of Plant Protection—National Research Institute, Węgorka 20, 60-318 Poznań, Poland
4 Department of Functional and Organic Food, Institute of Human Nutrition Sciences, Warsaw University of Life Sciences, 02-776 Warsaw, Poland
5 Department of Mathematical and Statistical Methods, Poznań University of Life Sciences, 28 Wojska Polskiego, 60-637 Poznań, Poland
6 Institute for Breath Research, University of Innsbruck, Innrain 66, A-6020 Innsbruck, Austria
* Correspondence: dariusz.piesik@pbs.edu.pl

**Abstract:** *Rumex confertus* is a rhizomatous, invasive, and difficult to control plant. Nevertheless, for sustainable agriculture, studies to biologically control *R. confertus* continue. Towards this, considerable attention has been devoted to investigating the emission of a wide array of volatile organic compounds (VOCs) from herbivore-damaged plants, which are known to induce protection measures in neighboring, undamaged plants. The goals of this study are to (1) determine if the profile of green leaf volatiles (GLVs), which are organic compounds naturally produced by undamaged plants, is similar to that provoked by the chemical stimulants Z-jasmone (ZJA) and dihydrojasmone (DJA), (2) establish if the *Apion miniatum* beetle's reproductive choices are influenced by their sex and mating status, and (3) assess if chemically stimulated GLV emissions can be used as signals to attract pests to *R. confertus* for biological control purposes. Synthetic forms of naturally produced Z-jasmone (ZJA) and dihydrojasmone (DJA), which can act as either an attractant or a repellent of various species of insects, were used to treat *R. confertus*. In olfactory analysis, the behavioral responses of individual insects to mated and unmated insects and to two GLV blends were evaluated. It was found that unmated male insects were fairly equally divided between going for food (Y-tube olfactometer arm with a GLV blend) and opportunities for reproduction (Y-tube olfactometer arm with unmated females). However, an increase in the GLV blend concentration attracted the insects. Meanwhile, unmated females were definitely indifferent to food and, independent of the GLV blend dose, were more interested in reproduction. Mated males, even with weak feed stimuli, increased their reproduction activity, in opposition to mated females. We provide evidence that synthetic GLV blends can be used to attract predators, making their use an effective means to biologically control *R. confertus*. The idea of applying synthetic GLVs as a biological control is based on the insects' mutual relationships, which work as follows: (i) mated males fully invade the weed, (ii) reproduction-driven females follow the mated males to R. *confertus*, and (iii) the unmated males follow the females with the purpose to reproduce. Therefore, all insect groups (mated and unmated males and females) can be induced to invade the weed. Upon feeding, the insects will damage the generative organs of *R. confertus*. We propose that the use of chemical stimulants to increase GLV emissions can be effectively used for weed (here *R. confertus*) control by attracting a plant pest (here *A. miniatum*).

**Keywords:** synthetic green leaf volatile stimulants; pest control; invasive plants; ecological chemistry; sustainable agriculture; Y-tube olfactometer

## 1. Introduction

The consequences of human activity in agriculture have been the introduction and naturalization of new plant species in many regions of the world. This in turn has increased the likelihood of new plant invasions, which brings negative consequences to the ecologies of endemic ecosystems. The uncontrolled spread of an invasive species poses a serious threat to biodiversity and for the efficient functioning of local ecosystems, the occurrence of which is increasing at an alarming rate worldwide [1]. The recognition of this serious problem is highlighted in the 8th article of the Biodiversity Convection, which is dedicated to the impact of invasive plants [2].

One of the world's most invasive plants is *Rumex confertus*. Of the potentially most invasive plant species in Europe, this plant has been categorized in the highest risk assessment, class III [2]. *R. confertus* is native to Europe and Asia [3], and has been reported as an invasive plant to the flora in Bulgaria [4], Latvia [5], Lithuania, Romania, Serbia, Hungary, and the Czech Republic [3]. Being rhizomatous, it is able to sprout again after physical and chemical management [3], and hence is difficult to control. It grows quickly, and produces a large number of viable seeds [6].

According to the 2016 USDA report [6], the genus *Rumex* comprises a number of species considered to be significant weeds. For example, *Rumex acetosella*, *R. conglomeratus*, *R. crispus*, and *R. obtusifolius* are weeds found in ruderal areas, and must be removed manually for effective control. *R. crispus* seeds are reported to be viable even after more than 80 years of vegetation in dry conditions. The plant is also toxic to poultry and causes mortality to sheep. *R. acetosella* causes high fever and dermatitis in humans and is also toxic to sheep. In contrast, the roots of many species belonging to the *Rumex* genus have been used in folk medicine since ancient times because of their gentle laxative effect [7]. With regard to the current study, *R. confertus* shares many of the abovementioned harmful traits, and hence is problematic if not kept under control [6].

Given the problems associated with the physical and chemical control of *R. confertus*, it is not surprising that biological control has been considered and implemented. One such control is associated with the soil type used. For example, it has been reported that Dystric Arenosol, Entic Podzol, Brunic Arenosol, and Calcaric Leptosol soils are unfavorable for *R. confertus* growth. It has also been found that *R. confertus* growth can be limited by reducing the available nitrogen and phosphorus in the soil [8]. Other aspects of biological control of *R. confertus* include the use of flowering plants or floral resources as a way to attract and retain an invasive plant's natural enemies for the protection of organic crop production. However, this approach has inconsistent outcomes given that there are few correlations between an increase in natural enemies and a reduction in plant damage [9]. According to Cloyd (2020) [9], the effect of plant diversity on the potential use of natural enemies to regulate insect pest populations below damaging levels is complex due to the plant–pest–natural enemy interaction. Therefore, plant diversity can either impair or promote the abundance of natural enemies and their ability to regulate insect pest populations and more importantly, reduce plant damage.

In nature, *R. confertus* is inhabited by a number of herbivorous insects, which while feeding will impair the plants' generative organs, with a high correlation between the plant damage index and the content of polyphenolic acids being confirmed [10]. Field and laboratory studies confirmed the potential of using *Gastroidea viridula* Deg. (Coleoptera: Chrysomelidae) as a biological control of *R. confertus* [11].

The interaction between damaged plants and the environment occurs by means of the emission of volatile organic compounds (VOCs), which provide signals to influence neighboring plants [12]. Thus, the VOCs released by plants have certain key ecological functions. They participate in plant-to-plant communication [13] and have an impact on the growth, disease resistance, dormancy, enzyme activity, photosynthesis, respiration, and the content of reactive oxygen species [12,14,15]. They can also be used to enhance weed biological control [16]. The largest group of such VOCs are terpenes, comprising isoprene, monoterpenes, and sesquiterpenes. Isoprene is associated with combatting abiotic stress,

while the mono- and sesqui-terpenes are primarily responsible for chemical plant-to-plant communication [13].

Plants react by sensing environmental stimuli. One of the most abundant groups of such chemical stimuli are plant-associated bacteria, which are part of the plant microbiome, comprising all microorganisms. Plants can acquire early warning information through their microbiome and plant–plant signals such as volatiles. The perception of VOCs generates intra- and inter-cellular signals that induce changes in a plant's physiology and metabolism [17]. In this way, the plant is connected to its plant-associated microbial community (microbiome), which has been shown to play an important role in plant-to-plant communication. However, a different set of VOCs is secreted by the plant during infection by pathogenic bacteria or fungi, and those volatiles play an important role in a plant's defense system [18]. Sensing the environmental stimuli is also a way for a plant to react to herbivores, with and without physical contact. Plants affected by a herbivore infestation produce VOCs, which induce microbiome adaptation in receiver plants, eliciting the defense response and improving the plant's fitness against stresses [17].

A plant's response to a pest attack is a subject of considerable interest in the agricultural industry [12,14,17,19]. Herbivore attack induces plants to produce specific herbivore-associated plant volatiles, that not only prime a plant's defense mechanism, but also influence the behavior of a plant's herbivores and natural enemies, by making them more susceptible or more resistant to parasitoids and pathogens [19]. The mutual interaction between plants and herbivores is a complex process involving constant modifications in the chemical responses between insects and plants.

Insects utilize plants for oviposition and as larval food sources in order to complete their life cycle. In response, plants utilize various mixtures of VOCs to improve their defense system and to help overcome the damages caused by the insects [20]. Plants use VOCs as a primary mechanism to repel rather than resist herbivore attacks [21]. In the event of an herbivore attack, plants defend themselves by using a set of VOCs specific for the particular herbivore. It has been shown, for example, that for the *Solanum tuberosum* (potato), the chewing Colorado potato beetle (*Leptinotarsa decemlineata*) and the piercing-sucking green peach aphid (*Myzus persicae*) pests induce clearly different volatile signatures [22]. Such studies prove that the VOC emissions influence plant resistance in the field and modulate the herbivore fitness, which can be used for pest control and ecology-based approaches in sustainable agriculture [19].

Within the VOCs there is a group of chemical compounds called the green leaf volatiles (GLVs) that are produced constantly at low levels by almost all unstressed plants [23], and immediately in high levels during mechanical damage and under biotic and abiotic stresses [14]. They also have direct toxic effects on bacteria and fungi, induce defense mechanisms of plants in the vicinity, and can be used to attract or repel herbivores and their natural enemies, which has been explored in detail by Ameye et al. [14]. An example of such studies is the use of the Y-tube olfactometer for testing the behavior of herbivores to various sets of VOCs released by plants because of herbivore attack, which are called herbivore-induced plant volatiles (HIPVs). Such blends have been thoroughly studied since they attract the herbivores' natural enemies, such as arthropod predators and parasitoids [24].

The ecological importance of VOCs is widely recognized, and the possibility of their use in crop protection has been the subject of debate for many years [25]. The reason for this long debate is, most probably, the irregularity of insect responses to odor interaction [26]. In recent studies, the importance of real-time odor environment has been highlighted [26]. Importantly, these show that the real-time odor environment contains various VOCs in different concentrations, which change constantly during the process of herbivores finding a host plant. Furthermore, bacteria and fungi emit a huge number and variety of VOCs. Microbial VOCs are composed of various chemical groups, including alcohols, ketones, terpenoids, alkenes, or sulfur-containing compounds. Unfortunately, their biological origins, activities, and functions are not well established or understood [27], which limits our knowledge to utilize them. This is compounded by our lack of understanding on the

level of complexity of interactions of plant VOCs with the environment. Hence, plant VOC studies attract significant interest, especially as VOCs emitted from plants can be used as a tool for the ecological control of pest insects [28].

Although great attention has been devoted to the study of the mechanism leading to the emission of VOCs by herbivore-damaged plants and the subsequent increase in the resistance to herbivore attack by neighboring undamaged plants [25,28,29], a key point that has not been studied thoroughly is the aspect of ecological specificity and context-dependency of the VOCs' influence on plants. This knowledge is crucial for deciphering the mechanisms governing the plant–plant and plant–insect communications, which will allow us to design effective crop-protection strategies [29]. In our study, we used the *Apion miniatum* insect, as it is a natural herbivore of *R. confertus*, which we have used successfully in an earlier study [30].

The objectives of this study are to (1) determine if the profile of green leaf volatiles (GLVs) provoked by ZJA and DJA application is similar to that induced by herbivore attack, (2) verify if the *Apion miniatum* reproductive choices are influenced by the insects' sex and mating status, and (3) assess if the chemical stimulants result in GLV emissions that attract hostile pests to *R. confertus* for its control.

## 2. Materials and Methods

To evoke an increased emission of GLVs, the *R. confertus* plants were subjected to Z-jasmone (ZJA) and dihydrojasmone (DJA) treatment. ZJA and DJA are organic compounds that are naturally produced by undamaged plants, including *R. confertus*, which can act as either attractants or repellents to various insects. Following treatment, the resulting induced GLVs were collected and identified by gas chromatography-mass spectrometry (GC-MS). Synthetic mixtures were then prepared to produce volatile blends that imitate plant GLVs. The synthetic blends were used to assess their influence on insect behavior. All steps of the methodology used in this experiment are presented below.

### 2.1. Plants and Z-Jasmone (ZJA) and Dihydrojasmone (DJA) Application

Plants of *R. confertus* were cultivated in a greenhouse (kept under light (16 h/day) and dark (8 h/day) conditions), with artificial, supplementary light and ambient humidity. The temperature of the greenhouse was adjusted according to the light and dark periods, namely $22 \pm 2\,^\circ$C and $18 \pm 2\,^\circ$C, respectively. After two months of cultivation as potted plants in steamed soil, the plants were randomly divided into groups, which were subjected to ZJA and DJA treatments. For either ZJA or DJA exposure, the stimulant compound was first diluted to give a quantity of 250 µg per 100 mL of distilled water. For the vapor exposure experiments, 250 µL of the stimulant solution was absorbed onto filter paper. The filter paper was then placed into a plastic centrifuge tube (2 mL) that was closed with a plastic lid. To facilitate constant ZJA or DJA evaporation, the lid was perforated with a 0.5 mm diameter needle. The aerial parts of the plants were placed individually into nalophan sleeves along with one plastic tube per plant.

### 2.2. Rumex Confertus Volatile Organic Compounds Collection and Analysis

The *R. confertus* GLVs, induced by application of the ZJA and DJA, were collected by placing the individual plant leaves into the Nalophan bags (odor- and taste-free cooking plastic bags), which were connected to a pump system (Thermo Fisher Scientific, Waltham, MA, USA). The air containing the GLVs was sucked into collector traps, which consisted of glass tubes (Analytical Research Systems, Inc., Gainesville, FL, USA) with a length of 76 mm and an outer diameter of 6.35 mm, with each tube containing 30 mg of Super-Q adsorbent (Alltech Associates, Inc., Deerfield, IL, USA). GLV collection was carried out for 3 h from four plants simultaneously, always between 10 am and 1 pm. To avoid the collection of any confounding volatiles coming from the outside, purified and humidified air was delivered to the Nalophan bag at a rate of 1.0 L min$^{-1}$, while suction to the trap was maintained at a rate of 0.8 L min$^{-1}$, i.e., a positive pressure was maintained inside the

Nalophan bags throughout. The collection of the VOCs released by the plants were started at 24 h, 48 h, and 72 h after the plants were exposed to ZJA or DJA and lasted for 3 h.

After collection, the GLVs were eluted from the adsorbent with 225 μL of hexane, which contained 7 ng of decane (Sigma-Aldrich, Steinheim, Germany) for use as an internal standard. Thereafter, a qualitative and quantitative analysis of the VOCs was performed using a GC AutoSystem XL (PerkinElmer, Shelton, CT, USA) equipped with a 30 m DB-5MS capillary column (0.25 mm internal diameter, 0.25-μm film thickness; Restek, Bellefonte, PA, USA). The injection temperature was kept constant at 250 °C and the GC program was set so that the temperature was raised from 40 to 200 °C at a rate of 5 °C min$^{-1}$. The MS parameters used were as follows: ion source temperature, 250 °C; interface temperature, 200 °C; and operating in scanning mode to cover 40–400 *m/z* using a scan speed of 1250 amu s$^{-1}$.

The identification of the GLVs was based on a comparison of experimentally obtained mass spectra and linear retention indices (calculated against the $C_8$–$C_{20}$ *n*-alkane mixture, Sigma-Aldrich, Steinheim, Germany) with those available in NIST 17 Mass Spectral and Retention Index Libraries (The National Institute of Standards and Technology, Gaithersburg, MD, USA). The identified GLVs quantification was based on the amount of added internal standard.

### 2.3. Insects

*Apion miniatum* Germ. (Coleoptera: Apionidae), of both sexes, were collected as adults from meadows located on the banks of the Vistula River (Bydgoszcz, Poland). Thereafter, the beetles were bred for an appropriate number of days on potted plants of *R. confertus* under controlled conditions; namely, at a temperature of 22 ± 2 °C and at a relative humidity of 60 ± 5%. For further experiments, only newly bred beetles of both sexes were used.

### 2.4. Preparation and Application of Green Leaf Volatiles Organic Compounds Blends

The volatile blends prepared were based on resulting GLV profiles emitted by *R. confertus* plants after ZJA or DJA treatment. Each blend was prepared in four concentrations for spraying at rates of 1, 10, 100, or 1000 ng min$^{-1}$. Briefly, the required amount of GLVs (Sigma-Aldrich, Steinheim, Germany) was diluted in 1 μL of hexane and thereafter diluted in 50 μL of hexane. The resulting blends were added onto filter paper, and then the carrier with a GLV blend was placed into a perforated microcentrifuge plastic tube and immediately subjected to olfactometry assays.

### 2.5. Evaluation of the Influence of Synthetic GLVs on Insects Behavior with the Use of Y-Tube Olfactometer

In order to evaluate the insects' behavior, Y-tube olfactometer (SYNTECH GmbH, Germany) was applied. One arm received the GLV synthetic blend using purified, humidified air as the carrier gas, while the second arm was used as a control by delivering to it just purified, humidified air with hexane. Both gases were flowed into the arms at a rate of 0.3 L min$^{-1}$. To avoid the accumulation of gases within the olfactometer, the vacuum pump sucked at 1.2 L min$^{-1}$ from the bottom. After each measurement, the system was washed using detergent, acetone, and hexane, and then kept at 80 °C for at least 1 h to eliminate traces of chemicals in the tube that would otherwise contaminate the next set of experiments.

For each GLV blend version and/or concentration, 20 *A. miniatum* adults of both sexes (separately) were tested. The beetles were divided into one of three groups during olfactory tests: (i) mated adult females and males; (ii) unmated adult females and males; and (iii) mated and unmated adult females and males. Observations were noted and beetle counting was carried out, until each individual insect had chosen one of the Y-tube olfactometer arms.

### 2.6. Statistical Analysis

The normality of the distribution of the amounts of the VOCs was tested using the Shapiro–Wilk normality test [30]. Multivariate analysis of variance was used to test multivariate differences between terms and doses of all the VOCs jointly. A two-way analysis of variance was carried out to determine the effects of doses and times as well as the dose × time interaction on the variability of particular VOCs. The mean values for VOC emission rates were calculated. Fisher's least significant differences (LSDs) were calculated for doses and times as well as the dose × time interaction and on this basis, homogeneous groups were determined. The relationships between particular VOCs were assessed using Pearson's correlation [31]. A canonical variance analysis was applied to present a multi-trait assessment of the similarity of the tested combinations of doses and times in a lower number of dimensions with the least possible loss of information. Mahalanobis distances [32] were suggested to measure the concentration distance for poly-VOC [33], the significance of which was verified using the critical value $D_\alpha$, the least significant distance [34]. Mahalanobis distances were calculated for all pairs of VOCs. All analyses were conducted using the GenStat 22nd edition statistical software package.

## 3. Results

### 3.1. Plant Volatile Emissions Caused by ZJA and DJA Applications

*R. confertus* exposed to either ZJA or DJA responded with the intense emission of GLVs. Following ZJA application, (Z)-3-hexenal (Z-3-HAL), (Z)-3-hexenol (Z-3-HOL), (Z)-3-hexenyl actetate (Z-3-HAC), Z-ocimene (Z-OCI), linalool (LIN), benzyl acetate (BAC), methyl salicylate (MAT), β-carophyllene (β-CAR), and β-farnesene (β-FAR) emissions were identified. In addition to these, following DJA application other volatile compounds were also observed to be released; namely, (E)-2-hexenal ((E)-2-HAL) and (E)-2-hexenol ((E)-2-HOL). The correlation results for ZJA and DJA stimulants are presented in Tables 1 and 2, respectively. All pairs of GLVs display positive correlation coefficients. The compounds and their amounts for reach treatment are given in the Supplementary Files (Table S1).

**Table 1.** Correlation coefficients between particular pairs of GLVs emitted from *R. confertus* after ZJA application.

| VOCs | Z-3-HAL | Z-3-HOL | Z-3-HAC | Z-OCI | LIN | BAC | MAT | β-CAR | β-FAR |
|---|---|---|---|---|---|---|---|---|---|
| Z-3-HAL | | | | | | | | | |
| Z-3-HOL | 0.87 [#] | | | | | | | | |
| Z-3-HAC | 0.87 | 0.87 | | | | | | | |
| Z-OCI | 0.76 | 0.77 | 0.79 | | | | | | |
| LIN | 0.70 | 0.79 | 0.80 | 0.90 | | | | | |
| BAC | 0.65 | 0.71 | 0.66 | 0.80 | 0.78 | | | | |
| MAT | 0.78 | 0.77 | 0.82 | 0.88 | 0.87 | 0.85 | | | |
| B-CAR | 0.79 | 0.80 | 0.82 | 0.91 | 0.90 | 0.87 | 0.92 | | |
| B-FAR | 0.72 | 0.73 | 0.76 | 0.88 | 0.82 | 0.85 | 0.89 | 0.88 | |

[#] All correlation coefficients were significant at the 0.001 level.

**Table 2.** Correlation coefficients between particular pairs of GLVs emitted from *R. confertus* after DJA application.

| VOCs | Z-3-HAL | E-2-HAL | Z-3-HOL | E-2-HOL | Z-3-HAC | Z-OCI | LIN | BAC | MAT | β-CAR | β-FAR |
|---|---|---|---|---|---|---|---|---|---|---|---|
| Z-3-HAL | | | | | | | | | | | |
| E-2-HAL | 0.93 [#] | | | | | | | | | | |
| Z-3-HOL | 0.89 | 0.90 | | | | | | | | | |
| E-2-HOL | 0.92 | 0.92 | 0.92 | | | | | | | | |
| Z-3-HAC | 0.95 | 0.93 | 0.88 | 0.90 | | | | | | | |
| Z-OCI | 0.79 | 0.79 | 0.84 | 0.79 | 0.77 | | | | | | |
| LIN | 0.80 | 0.77 | 0.78 | 0.78 | 0.77 | 0.91 | | | | | |
| BAC | 0.72 | 0.73 | 0.73 | 0.74 | 0.72 | 0.87 | 0.86 | | | | |
| MAT | 0.83 | 0.81 | 0.83 | 0.81 | 0.80 | 0.85 | 0.85 | 0.91 | | | |
| B-CAR | 0.84 | 0.84 | 0.84 | 0.84 | 0.82 | 0.89 | 0.86 | 0.89 | 0.90 | | |
| B-FAR | 0.74 | 0.76 | 0.79 | 0.77 | 0.73 | 0.88 | 0.86 | 0.88 | 0.86 | 0.94 | |

[#] All correlation coefficients were significant at the 0.001 level.

For both chemical stimulants, the highest levels of particular GLV emissions were observed in the 72 h measurements following a dose of 1000 ng min$^{-1}$. Nevertheless, the canonical correlation results (see Figure 1) show that the simultaneous influence of time and dose has a similar effect on the plant GLVs emission. An interesting observation was that a low dose of the stimulants (i.e., 10 ng min$^{-1}$) did not differentiate the GLV emission in time. On the other hand, the application of higher doses (i.e., 100 and 1000 ng min$^{-1}$) gave aligned results within the same time intervals.

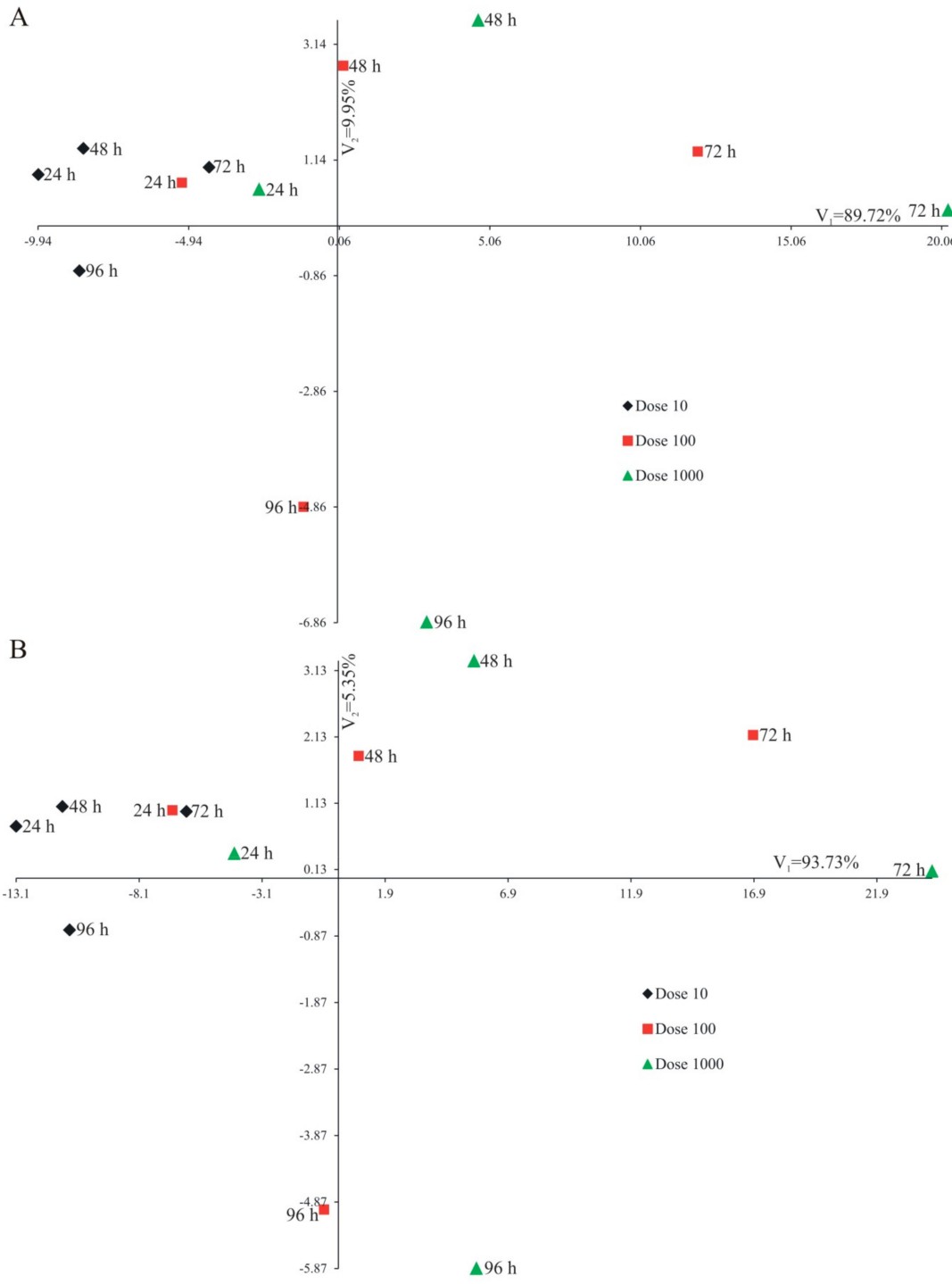

**Figure 1.** Results of a canonical correlation analysis of two parameters (time of GLV collection and stimulant doses) on VOCs emission levels, where (**A**) refers to ZJA and (**B**) refers to DJA application.

### 3.2. Apion Miniatum Behavior in Response to Feed or Reproduction Stimuli

The components of synthetic GLVs blends were selected based on the results of *R. confertus* plants treated with ZJA and DJA (see Tables 1 and 2). Next, the influence of synthetic GLV blends on the behavior of *A. miniatum* was evaluated in Y-tube olfactometer (Tables 3 and 4). It was observed that a separate application of lower dose blends (1 and 10 ng min$^{-1}$) attracted the insects, while the highest dose (1000 ng min$^{-1}$) was found to repel the insects (see Tables 3 and 4).

**Table 3.** The effect of a synthetic blend of three plant GLVs that are induced by ZJA on the number of *A. miniatum* adult females and males choosing to enter a Y-tube arm containing the blend odor or the control Y-tube arm with no odor, containing purified, humidified air and hexane solvent.

| Name of Mixed Compounds | Dosage ng min$^{-1}$ | No. of Females | | | No. of Males | | |
|---|---|---|---|---|---|---|---|
| | | + [2] | − [3] | $\chi^{2}$ [1] | + [2] | − [3] | $\chi^{2}$ [1] |
| (Z)-3-HAL | control (0.0) | 10 | 10 | 0.05 ns | 7 | 13 | 1.25 ns |
| + (Z)-3-HOL | 1 | 16 | 4 | 6.05 * (a) | 11 | 9 | 0.05 ns |
| + (Z)-3-HAC | 10 | 18 | 2 | 11.3 *** (a) | 16 | 4 | 6.05 * (a) |
| | 100 | 13 | 7 | 1.25 ns | 10 | 10 | 0.05 ns |
| | 1000 | 1 | 19 | 14.45 *** (r) | 2 | 18 | 11.3 *** (r) |

Legend for Tables 3 and 4: [1] level of significance (ns—not significant), (* $p < 0.05$), (** $p < 0.01$), (*** $p < 0.001$), (r)—repellent, (a)—attractant, [2] + Y—tube arm with tested amount of the compound, volatile diluted in hexane emitted from filter paper, [3] − Y—tube arm only with hexane emitted from filter paper.

**Table 4.** The effect of a synthetic blend of five plant GLVs that are induced by DJA on the number of *A. miniatum* adult females and males choosing to enter a Y-tube arm containing the blend odor or the control Y-tube arm with no odor containing purified, humidified air and hexane solvent.

| Name of Mixed Compounds | Dosage ng min$^{-1}$ | No. of Females | | | No. of Males | | |
|---|---|---|---|---|---|---|---|
| | | + [2] | − [3] | $\chi^{2}$ [1] | + [2] | − [3] | $\chi^{2}$ [1] |
| (Z)-3-HAL | control (0.0) | 11 | 9 | 0.05 ns | 9 | 11 | 0.05 ns |
| + (E)-2-HAL | 1 | 7 | 13 | 1.25 ns | 13 | 7 | 1.25 ns |
| + (Z)-3-HOL | 10 | 15 | 5 | 4.05 * (a) | 16 | 4 | 6.05 * (a) |
| + (E)-2-HOL | 100 | 17 | 3 | 8.45 ** (a) | 17 | 3 | 8.45 ** (a) |
| + (Z)-3-HAC | 1000 | 3 | 17 | 8.45 ** (r) | 3 | 17 | 8.45 ** (r) |

The attractant activity of the applied GLVs blends in light of the insects' sex and mating status was also evaluated (see Tables 5–8). It was observed that unmated males were fairly equally divided between their food interests (Y-tube olfactometer arm with GLV blend) and seeking a mate (Y-tube olfactometer arm with unmated females). Only a strong increase in the GLV blend concentration provoked food searching to be preferred above sexual activities. Meanwhile, unmated females were definitely indifferent to food stimulus, and, independent of the GLV blend dose (types and dose), they were more interested in reproduction opportunities (see Tables 5 and 6). The change in an insect's reproduction status resulted in an adjustment of the males' behavior, which was not observed for the female insects. *A. miniatum* mated males, even with weak stimuli for feeding, increased their reproduction activity, in opposition to the mated females, which were still more attracted to GLV blends, suggesting that they were more focused on feeding (see Tables 5 and 6).

**Table 5.** The effect of a synthetic blend of three GLVs (blend B1) on unmated beetles and on the number of *A. miniatum* adult females and males choosing to enter a Y-tube arm containing the blend odor or the control Y-tube arm containing purified, humidified air and hexane solvent (no odor). Only concentrations working as attractants (see Tables 1 and 2) were used.

| Name of Mixed Compounds after ZJA Application | Dosage ng min$^{-1}$ | No. of Unmated Females | | | No. of Unmated Males | | |
|---|---|---|---|---|---|---|---|
| | | + [2] B1 | _ [3] Unmated Males | $\chi^{2\,[1]}$ | + [2] B1 | _ [3] Unmated Females | $\chi^{2\,[1]}$ |
| (Z)-3-HAL | control (0.0) | 11 | 9 | 0.05 ns | 10 | 10 | 0.05 ns |
| + (Z)-3-HOL | 1 | 9 | 11 | 0.05 ns | 4 | 16 | 6.05 * |
| + (Z)-3-HAC | 10 | 15 | 5 | 4.05 * | 2 | 18 | 11.3 *** |

Legend for Tables 5–8: [1] level of significance (ns—not significant), (* $p < 0.05$), (** $p < 0.01$), (*** $p < 0.001$), [2] + Y—tube arm with tested mixture of B1 or B2, volatile diluted in hexane emitted from filter paper, [3] − Y—tube arm with unmated/mated females or males, B1 and B2—two blends of attractants (only), where B1 is composed of (Z)-3-HAL, (Z)-3-HOL, and (Z)-3-HAC, and B2 is composed of (Z)-3-HAL, (E)-2-HAL, (Z)-3-HOL, (E)-2-HOL, and (Z)-3-HAC.

**Table 6.** The effect of a synthetic blend of three GLVs (blend B1) on mated beetles and on the number of *A. miniatum* adult females and males choosing to enter a Y-tube arm containing the blend odor or the control Y-tube arm containing purified, humidified air and hexane solvent (no odor). Only concentrations working as attractants (see Tables 1 and 2) were used.

| Name of Mixed Compounds after ZJA Application | Dosage ng min$^{-1}$ | No. of Mated Females | | | No. of Mated Males | | |
|---|---|---|---|---|---|---|---|
| | | + [2] B1 | − [3] Mated Males | $\chi^{2\,[1]}$ | + [2] B1 | − [3] Mated Females | $\chi^{2\,[1]}$ |
| (Z)-3-HAL | control (0.0) | 11 | 9 | 0.05 ns | 9 | 11 | 0.05 ns |
| + (Z)-3-HOL | 1 | 15 | 5 | 4.05 * | 4 | 16 | 6.05 * |
| + (Z)-3-HAC | 10 | 17 | 3 | 8.45 ** | 3 | 17 | 8.45 ** |

**Table 7.** The effect of a synthetic blend of three GLVs (blend B2) on unmated beetles and on the number of A. miniatum adult females and males choosing to enter a Y-tube arm containing the blend odor or the control Y-tube arm containing purified, humidified air and hexane solvent (no odor). Only concentrations working as attractants (see Tables 1 and 2) were used.

| Name of Mixed Compounds after DJA Application | Dosage ng min$^{-1}$ | No. of Unmated Females | | | No. of Unmated Males | | |
|---|---|---|---|---|---|---|---|
| | | + [2] B2 | _ [3] Unmated Males | $\chi^{2\,[1]}$ | + [2] B2 | _ [3] Unmated Females | $\chi^{2\,[1]}$ |
| (Z)-3-HAL | control (0.0) | 13 | 7 | 1.25 ns | 11 | 9 | 0.05 ns |
| + (E)-2-HAL | 1 | 17 | 3 | 8.45 ** | 2 | 18 | 11.3 *** |
| + (Z)-3-HOL | 100 | 18 | 2 | 11.3 *** | 0 | 20 | 18.5 *** |
| + (E)-2-HOL | | | | | | | |
| + (Z)-3-HAC | | | | | | | |

**Table 8.** The effect of a synthetic blend of three GLVs (blend B2) on mated beetles and on the number of A. miniatum adult females and males choosing to enter a Y-tube arm containing the blend odor or the control Y-tube arm containing purified, humidified air and hexane solvent (no odor). Only concentrations working as attractants (see Tables 1 and 2) were used.

| Name of Mixed Compounds after Dihydrojasmone Application | Dosage ng min$^{-1}$ | No. of Mated Females | | | No. of Mated Males | | |
|---|---|---|---|---|---|---|---|
| | | + $^{(2)}$ B2 | – $^{(3)}$ Mated Males | $\chi^{2\,(1)}$ | + $^{(2)}$ B2 | – $^{(3)}$ Mated Females | $\chi^{2\,(1)}$ |
| (Z)-3-HAL | control (0.0) | 13 | 7 | 1.25 ns | 11 | 9 | 0.05 ns |
| + (E)-2-HAL | 1 | 16 | 4 | 6.05 * | 4 | 16 | 6.05 * |
| + (Z)-3-HOL | 100 | 20 | 0 | 18.5 *** | 2 | 18 | 11.3 *** |
| + (E)-2-HOL | | | | | | | |
| + (Z)-3-HAC | | | | | | | |

For GLV blends, based on DJA application to the plants (see Tables 7 and 8), the behavior of mated and unmated males, as well as mated and unmated females, was consistent with the results obtained for blend B1. Females, independent of their mating status, were still more attracted to the possibility of feeding. Insect males were not affected by GLV blends, even with very high intensities, but they were affected by exposure to the unmated females (Tables 7 and 8).

## 4. Discussion

Plant responses to pest attacks are the subject of considerable interest [12,14,17,19]. A herbivore attack stimulates plants to produce and emit herbivore-specific volatiles that, among others, influence the behavior of the herbivores [18]. Various aspects of plant VOCs have been investigated, for instance, the use of kairiomones, chemicals that mediate interspecific interactions beneficial to organisms that detect it. It has been noted that kairiomones attract the natural enemies of a plant's pest, which has been used to design effective plant control strategies [35]. Another example is the use of VOCs to specifically control hyperparasitoids that can jeopardize pest control by affecting the populations of their parasitoid hosts, which leads to an outbreak of pests [36]. In a previous study, a specific single leaf VOC induction on *R. confertus* was shown to be dose-dependent for herbivore *Gastrophysa polygoni* attraction/repellence to individual compounds [37]. Additionally, it has been shown that *A. miniatum* herbivory on the *R. confertus* induces VOCs that alter the insects' orientation responses [30]. Furthermore, it has been reported that *Gastrophysa viridula* and *G. polygoni* (Coleoptera: Chrysomelidae) beetles respond to a blend of synthetic VOCs [38].

In this study, we exploited the observation that VOCs can influence pest behavior. However, instead of using plant VOCs directly, we used synthetic blends of *R. confertus* GLVs. We have shown that the GLVs produced by applying stimulants are consistent with earlier findings for *R. confertus* when *Hypera rumicis* pests influence was investigated [39]. However, some important differences were observed. For example, the two chemical stimulants used in this study, ZJA and DJA, did not cause the emission of geranyl acetate, methyl anthranilate, or indole. According to a review by Turlings and Erb [25], the reason for this may be associated with the lack of any physical damage to the plant's leaves.

Given that *R. confertus* is an invasive plant throughout Central Europe [3], the potential use of GLVs to attract herbivores, especially belonging to *A. miniatum*, is a promising possibility for use in natural weed control [40]. Therefore, we propose that the application of stimulants, which cause GLV emissions, could be used as a natural way to limit *Rumex* sp. growth. However, some risks and limitations have to be considered. First, high doses of

GLV blends result in them becoming a repellent. Thus, limits of the dose of stimulants need be precisely determined to avoid this. Furthermore, the attraction of herbivores based only on GLVs is limited without supporting information on herbivore sex pheromones [16]. Our results confirm this, since *A. miniatum* were always attracted to sources of food, independent of their mating status, while males were more focused on reproduction, based on their choices in the Y-tube experiment we performed (Tables 5–8). Nevertheless, the differences in the behavior of males and females suggest that the strategy of natural *R. confertus* control by *A. miniatum*, who are attracted to the GLVs that are emitted by plants, does not need to be supplemented with sex pheromones, because mated females would invade the weed just to feed, while males will follow them due to their desire to reproduce.

To summarize, we have provided evidence that shows that synthetic GLV blends can be used as an effective natural control of *R. confertus*. This is based on the insects' mutual relationships, which works as follows: the mated females fully invade the weed and both mated and unmated males will follow females in order to reproduce. This means that all insects groups invade the weed, which upon feeding damages the generative organs of *R. confertus*.

## 5. Conclusions

The artificial increase in *R. confertus* green leaf volatile emissions has the potential of acting as an effective natural tool for the attraction of *A. miniatum.* This study suggests that the application of ZJA or DJA as chemical stimulants should cause the emission of green leaf volatiles at the rates of 1–10 ng min$^{-1}$ or 1–100 ng min$^{-1}$, respectively. Furthermore, we have shown that *A. miniatum* attraction to the plant is dependent on the insects' sex and mating status. While female insects were found to be only interested in the search for food after mating, males were more interested in finding a mate, independent of the females' mating status. Nevertheless, the application of ZJA or DJA will still be effective, because of the insects' mutually dependent behavior; namely, the mated females invading the weed for food will be followed by males (mated and unmated) with their desire for reproduction ahead of that of food.

In conclusion, we propose that the use of chemical stimulants to increase GLV emissions could be used as an effective control of weeds through their use to attract a natural pest.

**Supplementary Materials:** The following supporting information can be downloaded at: https://www.mdpi.com/article/10.3390/app13042253/s1, Table S1: The VOCs emitted from *R. confertus* after ZJA and DJA application. The amounts of VOCs for each treatment are given as a mean along with its standard deviation (s.d.).

**Author Contributions:** Conceptualization, D.P.; methodology, D.P., J.B.; software, D.P., J.B. and V.R.; validation, J.Ł., K.K. and J.B.; formal analysis, J.B. and M.G.; investigation, D.P., V.R. and M.G.; resources, D.P. and M.G.; data curation, D.P.; writing—original draft preparation, J.Ł. and K.K.; writing—review and editing, D.P., V.R. and C.A.M.; visualization, D.P. and J.B.; supervision, D.P.; project administration, D.P.; funding acquisition, D.P. All authors have read and agreed to the published version of the manuscript.

**Funding:** This research received no external funding.

**Institutional Review Board Statement:** Ethical review and approval were waived for this study due to the lack of legal demands for using the insects in the scientific studies, in Poland.

**Informed Consent Statement:** Not applicable.

**Data Availability Statement:** The data supporting reported results can be found in the Table S1 provided in the Supplementary File. If needed more details will be provided on demand.

**Conflicts of Interest:** The authors declare no conflict of interest within the work reported in this paper.

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
