# Peer review of "Green Leaf Volatile Function in Both the Natural Defense System of Rumex confertus and Associated Insects’ Behavior"

_applsci, doi:10.3390/app13042253_

Round 1
Reviewer 1 Report
Dear authors,
Your results are poorly described and do not reflect your conclusions. I recommend you rewrite your manuscript with careful detailing of the results you obtained and the methods you used. Explain carefully why you did what you did and what result you got. Then you can discuss how these results are giving a nice insight on the plant-insect relationship and how you can exploit it for pest management. Curently the results do not match the conclusions, and many results are not shown which you should have gotten (based on the method you described) and should be very interesting. Review your manuscript to revise the use of English grammar so that the sentences are well structured and the message you deliver is the one you want to deliver.
see below more detail examples but you must revise everything.
Best of luck!
examples:
-l19 : "Rc, A rhizomatous invasive plant, IS difficult to control. "
-The two first sentences do not make much sense.
Please rewrite the abstract to introduce the concept with simpler terms and make a clear story between the background, your research questions and the method used.
you are talking about R confertus and then of A. miniatum without first introducing what is what. People that are new to the field will not understand which is the plant and which is the pest from the latin name.
why use the SjA and DjA? this should be clear from the abstract.
l34-38 How do you deduct the second sentence from the first?
Introduction:
as above, review the writing style. Pay attention to the logical order of your arguments.
l86 why talk about this species and not about the one you study?
l91 whytalk about allelopathy here? how is this relevant to understand your study and its impact?
I think you go in too much details about plant chemistry and not enough on the pest you study and the interaction between the two.
you did not introduce A. miniatum at all?!
material and methods:
l 169 why use these products?
l 174 how do the plants survive in sterilized soil?
l 179 what base for the quantity of products used? the method used?
l 194 why the time restriction?
l 221 how did you decide which compounds from the released headspace were categorized as GLV?
l238 not a memory effect but to avoid traces of chemicals to remain in the tube and contaminate the next experiments
l246: what is the distribution of the VOCs? do you mean the amounts produced? the presence of each?...?
Results
I am missing the qualitative and quantitative result here: where are the compounds and their amounts for each treatment? they should be presented before to compare the two?
l277 which are these particular GLV?
table 3 and associated results: which are the components of the blends and their ratios?
The tables are difficult to read. You could use whole words instead of symbols in the headers. Keep the symbols for the statistical results.
l315 how do you know the females were focused on reproduction?
Discussion:
l342 Kairomones
How is your results supporting your claims? I am failing to find the resulting outcome of this study: what was the purpuse of stimulating with Jasmonate compounds? how is the behavioural change you observed interesting and useful? which compounds were actually relevant? can they be used ?
l371 this is a false claim: you cannot pretend to know what the insect is focused on based on whether they were attracted or not by the chemicals.
l376 which blend of what chemicals? how is it a natural control? why?
Author Response
Response to the Reviewer 1

Reviewer 2 Report
Abstract
Line 22- “herbivore-damaged plants” what plant are you referring to? Rumex confertus?
Lines “19-22” and “23-28” don’t have flow. I had a hard time following and understanding the connection between the background and the goals.
Introduction
It is worth mentioning that in some cases Rumex is used for folk medicine.
Lines 81-82 “few correlations between an increase in natural enemies with a reduction in plant damage”- please consider elaborating this further.
Lines 99-101 “One of the most abundant groups of such chemical stimuli are plant-associated bacteria, called microbiome.” Microbiome is not only specific to bacteria.
Line 101 “Perception of such VOCs” are you referring to VOCs associated to microbiome activities?
Materials and Methods
Lines 179-181- “For either ZJA or DJA exposure, the stimulant compound was first diluted to give a quantity of 250 µg per 100 mL of distilled water. For the vapour exposure experiments, 200 µL of the stimulant solution was absorbed onto filter paper.” Were there two types of exposures? I did not follow you here.
Line 87 “application of the ZJA and DJA” was this application similar what was done above in section 2.1?
Results
Table 1 & 2: Italicize R. confertus
Table 3, 4,5 & 6: Italicize A. miniatum
Lines 304-307: the case of male preference between food and mate please include how it was done in the method section.
The first objective of the study “(1) determine if the profile of GLVs provoked by chemical stimulants is similar to that induced by herbivore attack” was not addressed in both the methods and no related result was reported in the result section.
Lines 340-341 “kairomones” are interspecific, not intraspecific
Author Response
Response to the Reviewer 2

Reviewer 3 Report
· In line 68, page 2: “The seeds of R. crispus seed are reported to be viable after 80 years of vegetation in dry conditions”, you wrote seed twice.
· In line 123, page 3: “It has been shown, for example, that for the Solanum tuberosum potato, the chewing Colorado Potato Beetle …….” It will be better to put potato between brackets ( ).
· What is the concentrations of each compounds emitted by this plants in response to ZJA and DJA stimulation? If not known how did you chose the concentration of each compounds compounds in the mixture?
· In 3.1. Plant volatiles emission caused by ZJA and DJA applications part you stated that 9 molecules and 11 molecules have been emitted by plant in response to ZJA and DJA stimulation respectively, However, to evaluate the effect of synthetic blend preparation on insect behaviour, you have used just, three GLVs for ZJA and five GLVs for DJA. Why did test other molecules which may influence the insect behaviour even at low concentrations.
· In all parts of manuscripts, you stated that “ A. miniatum males were always attracted to sources of food, independent of their mating status, while females were more focused on reproduction” the same conclusion have been reported in line 305 to 314, and the same thing in conclusion section. However, the results reported in results in tables 5 and 6 show the opposite, means that “ Males were more interested in reproduction opportunities independent of the GLV blend dose and insect’s reproduction status”.
· There are no tables 7 and 8 as you said in line 336 page 10.
· How did you choose insect?
Author Response
Response to the Reviewer 3

Round 2
Reviewer 1 Report
Thank you for addressing my comments, the manuscript is better now.
Reviewer 3 Report
The improved manuscript can be accepted for publication